# Unsupervised learning of the brain connectivity dynamic using residual D-net

**Youngjoo Seo**[1], **Manuel Morante**[2,3], **Yannis Kopsinis**[4,3] **and Sergios Theodoridis**[5,2,3]
[1]Signal Processing Laboratory 2, EPFL (Switzerland)
[2]Dept. of Informatics and Telecommunications, University of Athens (Greece)
[3]Computer Technology Institute & Press "Diophantus" (CTI), Patras (Greece)
[4]LIBRA MLI Ltd, Edinburgh (UK)
[5]IAASARS, National Observatory of Athens, GR-15236, Penteli (Greece)
youngjoo.seo@epfl.ch, morante@cti.gr, kopsinis@ieee.org, stheodor@di.uoa.gr

## Abstract

In this paper, we propose a novel unsupervised learning method to learn the brain dynamics using a deep learning architecture named residual D-net. As it is often the case in medical research, in contrast to typical deep learning tasks, the size of the resting-state functional Magnetic Resonance Image (rs-fMRI) datasets for training is limited. Thus, the available data should be very efficiently used to learn the complex patterns underneath the brain connectivity dynamics. To address this issue, we use residual connections to alleviate the training complexity through recurrent multi-scale representation. We conduct two classification tasks to differentiate early and late stage Mild Cognitive Impairment (MCI) from Normal healthy Control (NC) subjects. The experiments verify that our proposed residual D-net indeed learns the brain connectivity dynamics, leading to significantly higher classification accuracy compared to previously published techniques.

## 1 Introduction

Alzheimer's Disease (AD) is the most common degenerative brain disease associated with dementia in elder people [1], and it is characterized by a progressive decline of memory, language and cognitive skills. The transition from cognitive health to dementia flows throw different stages, and it may require decades until the damage is noticeable [2].

Unfortunately, the precise biological mechanisms behind the AD remain unknown, to a large extent, and this makes the development of an effective treatment difficult. Moreover, the costs of Alzheimer's care constitutes a substantial burden on families, which exacerbates through the evolution of the disease [3]. For these reasons, early detection is crucial to prevent, slow down and, hopefully, stop the development of the AD.

Towards this goal, several studies point out that an intermediate stage of cognitive brain dysfunction, referred as Mild Cognitive Impairment (MCI), is a potential precursor of AD [3] (especially with respect to memory problems, referred as amnesic MCI). Although the final transition from MCI to AD varies per individual, a recent systematic review of 32 available studies reported that at least 3 out 10 patients with MCI developed the AD over the period of five or more years.

During the early stages of the AD and MCI, the brain operates so that to allow the individuals to function normally by inducing abnormal neuronal activity, that compensates for the progressive loss of neurons. These fluctuations can be measured using rs-fMRI, which is a powerful non-invasive technique to examine the brain behavior. Therefore, the rs-fMRI provides valuable information that

1st Conference on Medical Imaging with Deep Learning (MIDL 2018), Amsterdam, The Netherlands.

Table 1: Demographics of the healthy control subjects (NC), patients with eMCI and patients with LMCI

|  | NC | eMCI | LMCI |
|---|---|---|---|
| Number of Subjects | 36 | 31 | 26 |
| Male/Female | 14/22 | 15/16 | 15/11 |
| Number of Scans | 100 | 100 | 77 |
| Male/Female | 37/63 | 58/42 | 41/36 |
| Age (mean±SD) | 72.7±4.5 | 72.4±3.8 | 74.3±3.4 |

allows to study the brain connectivity dynamics and, potentially, to detect individuals with AD or MCI from healthy subjects.

Nowadays, several methods have been proposed to classify subjects with MCI from healthy subjects using fMRI data [4]. The most basic approach consists of a direct study of the mean Functional Connectivity (FC). For example, features from the FC matrix [5] or graph theoretical approach [6] are proposed to perform the classification task. However, two practical limitations restrict these approaches: first, the manual feature designing requires an extensive domain knowledge of the brain connectivity dynamics and, second, the limited number of the available data samples makes it difficult to find a proper model that will generalize in different datasets.

On the other hand, a more sophisticated approach is proposed in [7] to address these two problems: this method automatically learns the features from the data using a Deep Auto-Encoder (DAE) by avoiding potential human biases. Nevertheless, the DAE does not consider any information regarding the brain connectivity dynamics, which is crucial to understand the AD.

Accordingly, any alternative deep learning method must simultaneously consider the structure and the dynamics of the brain functional connectivity, for automatically extracting significant features from the data. However, since complex deep learning architectures usually require a large number of training samples, the lack of sufficient data constitutes the major practical limitation of such methods.

For all these reasons, in this paper, we introduce a recurrent multi-scale deep neuronal network, named residual D-net, to analyze the brain behavior. The main novelty of the presented architecture is that it allows us to unravel the brain connectivity dynamics, but, efficiently learning with a limited number of samples, which constitutes the most common scenario in practice.

Therefore, we applied our proposed residual D-net to learn the brain connectivity dynamics of our subjects. Then, we feed the learned brain dynamic features into a classifier to distinguish subjects with MCI from healthy individuals.

## 2 Materials and Preprocessing

In this study, we use a public rs-fMRI cohort from the Clinical Core of Alzheimer's Disease Neuroimaging Initiative (ADNI)[1], which has established a competitive collaboration among academia and industry investigation focused on the early identification and intervention of AD [8].

Among the different datasets of ADNI (including the latest studies *ADNI go* and *ADNI 2*), there are data sets referring to patient with early stage of Mild Cognitive Impairment (eMCI), and patients with an advanced stage of the condition referred as Late stage Mild Cognitive Impairment (LMCI). In this paper, we report studies for both datasets separately.

### 2.1 ADNI Cohorts

The final used cohort comprises 277 scans from 36 Normal healthy Control (NC) subjects, 31 patients with eMCI and 26 patients with LMCI (see Table (1)). We distinguish between scans and subjects because some subjects have several scans at different points; the same person has undergone the scan at different times. This consideration is crucial: otherwise, we can introduce potential bias that affects the accuracy of the method.

---

[1]Availiable at `http://adni.loni.usc.edu/`

With respect to the data acquisition, all the rs-fMRI scans were collected at different medical centers using a 3 Tesla Philips scanners following the same acquisition protocol [9]: Repetition Time (TR) = 3000 ms, Echo Time (TE) = 30 ms, flip angle = 80°, matrix size 64×64, number of slices = 48 and voxel thickness = 3.313 mm. Each scan was performed during 7 minutes producing a total number of 140 brain volumes.

## 2.2   Preprocessing

The functional images were preprocessed using the Data Processing Assistant for Resting-State fMRI (DPARSF) toolbox[2] and the SPM 12 package[3] following standard preprocessing steps:

   – First, we discarded the first 10 volumes of each scan to avoid T1 equilibrium effects and we applied a slice-timing correction to the slice collected at TR/2 to minimize T1 equilibrium errors across each TR.

   – After correcting the acquisition time, we realigned each time-series using a six-parameter rigid-body spatial transformation to compensate for head movements [10]. During this step, we excluded any scanner that exhibited a movement or rotation in any direction bigger than 3mm or 3° respectively.

   – Then, we normalized the corrected images over the Montreal Neurological Institute (MNI) space and resampled to 3 mm isotropic voxels. The resulted images were detrended in time through a linear approximation and spatially smoothed using a Gaussian filtering with FWHM = 4 mm.

   – Finally, we removed the nuisance covariates of the white matter and the cerebrospinal fluid to avoid further effects and focused on the signal of the grey matter, and we band-pass filtered (0.01-0.08 Hz) the remaining signals to reduce the effects of motion and non-neuronal activity fluctuations.

## 2.3   Brain network analysis

In order to investigate the behavior of the brain functional connectivity, we labeled each brain volume into 116 Regions of Interest (ROIs), using the Automated Anatomical Labeling (AAL) atlas[4]. This atlas divides the brain into macroscopic brain structures: 45 ROIs for each hemisphere and 26 cerebellar ROIs. In this study, we excluded the 26 cerebellar ROIs, because theses areas are mainly related to motor and cognitive functional networks [11].

Then, we estimated a representative time course by averaging the intensity of all the voxel within each ROI, and we normalized the values in the range -1 to 1. Finally, we folded all the time courses into a matrix $\mathbf{R} \in \mathbb{R}^{90 \times 130}$, where each row contains the time evolution of one specific ROI.

# 3   Proposed methods

In this paper, we propose a novel residual D-net framework to model the brain connectivity dynamics. First, the selective brain functional connectivity dynamics, used as input for the residual D-net is presented. Then, the details of residual D-net will be described.

## 3.1   Selective Brain Functional Connectivity Dynamics

In order to capture the brain connective dynamics in the rs-fMRI, we examine the time-varying functional connectivity (FC) variability via windowing correlation matrices [12], which provides a fair estimate of the natural dynamics of the functional brain connectivity.

However, our goal is to identify individuals that will potentially develop AD. Consequently, we restricted our study of the whole-brain dynamics to just a few areas that may suffer damage due to the AD, which, also, reduces the pattern complexity of the brain functional connectivity.

---

[2]DPARSF: Available at `http://rfmri.org/DPARSF`
[3]SPM 12: Available at `http://www.fil.ion.ucl.ac.uk/spm`
[4]AAL documentation available at `http://www.gin.cnrs.fr/en/tools/aal-aal2/`

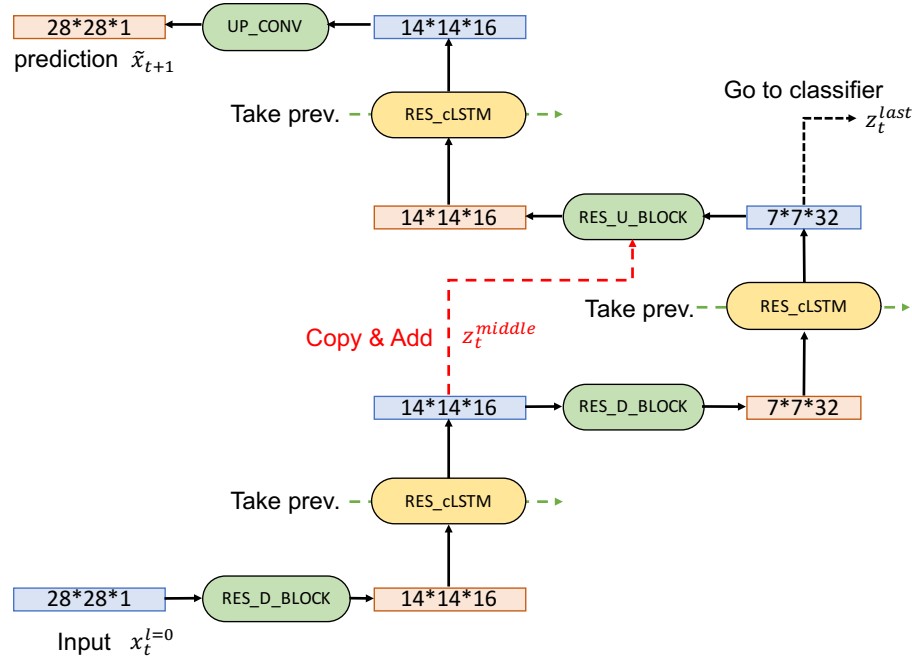

Figure 1: Residual D-net architecture

In this way, severals studies have pointed that certain brain areas are more likely to be affected by the AD. These areas are localized in the Frontal Lobe [13], the Hypocampus [14] and the Temporal Lobe [14], [15] . Therefore, we limited our study of the brain connectivity dynamics to 28 ROIs that are vulnerable to AD.

Thus, using this specific set of ROIs and following the method described in [12], for each scan ($\mathbf{R}_i$), with $i = 1, 2, \ldots, N$, where $N$ is the total number of analyzed scans, we estimated the dynamic FC through a sliding window approach, and we computed each covariance matrix from a windowed segment of $\mathbf{R}_i$. We applied a tapered window created by convolving a rectangle (with = 10 TRs=30) with a Gaussian ($\sigma = 4$TRs) and a sliding window in steps of 2 TRs, resulting in a total number of 56 windows.

Accordingly, the result of each scan contains a sequence of 56 covariance matrices that encode the connectivity dynamics of the studied ROIs. These sequential matrices comprise the FC dynamics of the 28 ROIs, and we will use them as an input to the proposed method. Figure (4.a) shows examples of input sequences of these covariance matrices.

### 3.2 Residual D-net

The proposed model needs to understand the dynamics of brain FC; that is, how the pattern within the covariance matrix changes along time. Furthermore, the model should be very efficient to learn the dynamics given a limited number of training data.

To address these issues, the proposed residual D-net has three major properties that allow the model to learn with relatively few training samples, while retaining its capacity to learn complex dynamics. Figure (1) shows the main architecture of the proposed residual D-net, which is formed by three main components: up residual block, down residual block (RES_U/D_Block) and a residual convolutional long short-term memory block (RES_cLSTM).

**RES_U/D_Block:**  The residual network (resNet) [16] is a competitive deep architecture capable to produce a detailed decomposition of the input data. The residual connection in the resNet constrains the network to learn a residual representation, so that to facilitate the training. We exploit this property to learn complex patterns in the input, while keeping the training to be simple. In addition, we add an "average pooling" layer and "up convolution" layer, to express the multi-scale representation. The

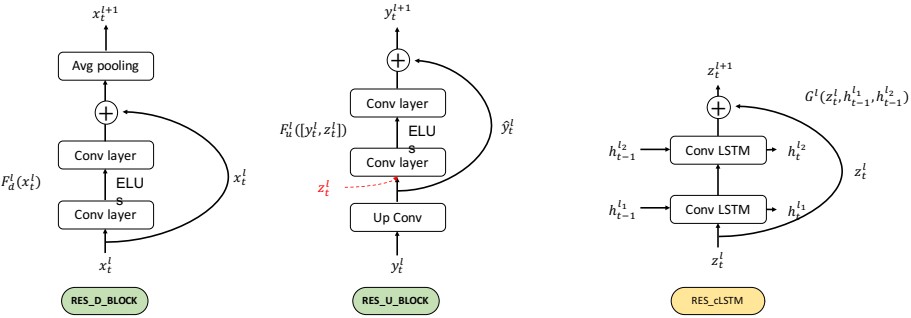

Figure 2: Major component of the residual D-net: Donw/up Residual Block and Residual convLSTM Block

formulations of each down/up residual block can be expressed as follows:

$$x_t^{l+1} = \text{avgpool}(F_d^l(x_t^l) + x_t^l), \tag{1}$$

$$y_t^{l+1} = F_u^l([\hat{y}_t^l, z_t^l]) + y_t^l, \tag{2}$$

where $x_t^l$ and $y_t^l$ are the inputs of the $l_{th}$ RES_D/U_Block, respectively. Each block has a bypass identity connection to fit the residual mapping from the input. We denote each convolutional layer in the block as $F_d^l(x_t^l)$ and $F_u^l(y_t^l)$ in Figure (2), which are composed of two $3 \times 3$ convolutional layers and we employ the exponential linear units (ELUs) [17] as the nonlinear activation function. The major difference lies in in their "up/down sampling" layer. In the RES_D_BLOCK, a average pooling layer is attached to down sample the input. In the RES_U_BLOCK, we use a up-conv layer for up-sampling ($\hat{y}_t^l$) the input and it is concatenated with $z_t^l$, which comes from the high resolution feature map in the upper RES_cLSTM Block.

**RES_cLSTM Block:** The convolutional LSTM [18] is a well-known Recurrent Neural Network (RNN) model, capable of capturing spatial-temporal features in a video sequence. As we described above, the brain dynamics is represented as a sequence of images. Thus, the use of a convolutional LSTM is fully justified by the nature of our task. Moreover, the use of the residual connection, together with the convolutional LSTM, facilitates the training, while retaining the spatial-temporal information. The connection was designed in a way similar to existing residual LSTM models [19, 20, 21] with two concatenated LSTM blocks with identity connection as shown in Figure (2). The formulation of the Residual Convolutional LSTM block can be expressed as follows:

$$z_t^{l+1} = h_t^{l_2} + z_t^l, \tag{3}$$

$$h_t^{l_2} = G^l(z_t^l, h_{t-1}^{l_1}, h_{t-1}^{l_2}). \tag{4}$$

Here, $z_t^l$ is the input of the $l_{th}$ RES_cLSTM Block and $h_{t-1}^{l_1}$, $h_{t-1}^{l_2}$ represents hidden states of the convolutional LSTM layer from previous $t-1$ time step. The function $G^l(z_t^l, h_{t-1}^{l_1}, h_{t-1}^{l_2})$ represents the $l_{th}$ two-layered convolutional LSTM that maps dynamics of the input pattern into the current hidden states($H_t^l : [h_t^{l_1}, h_t^{l_2}]$). Similarly to the residual block, all convolutional layer uses $3 \times 3$ size filter.

**Structure of residual D-net:** Using the residual blocks as components, we build a 2-depth U-net architecture for multi-scale representation. The U-net framework [22] was developed for dealing with deep representative learning tasks with few training samples. We adopt the same framework to take advantage of the rich feature representation and the efficient learning scheme. In addition, we add a recurrent flow to capture the dynamic behavior, so that the architecture forms **D-shape**.

As shown in Figure (1), The input $x_t^{l=0}$ comprises $28 \times 28$ images of the correlation map at $t$ time step. RES_D_Block decreases the input size by half and increases the feature map by two starting from the initial 16-feature map size. The feature maps are contracting until they reach the last RES_cLSTM block. These abstract embeddings ($z_t^{last}$) are finally used later on for the classification. During the expansion path, the feature map from the middle-depth layer, $z_t^{middle}$, is concatenated via a skip-connection. This multi-scale way of training allows to learn the complex patterns of the input sequences and to capture the dynamic changes in the hidden state of the convolutional LSTM.

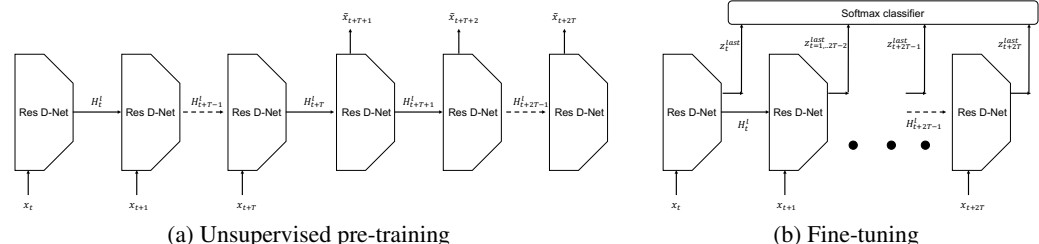

(a) Unsupervised pre-training          (b) Fine-tuning

Figure 3: Training scheme of the (a) unsupervised pre-training and (b) supervised fine-tuning using the residual D-net.

### 3.3 Unsupervised pre-training and fine-tuning

First, we train our residual D-net with sequences of correlation maps by predicting a few steps ahead of the sequences. Given $T$-time step input sequences, residual D-net predicts the output until next $2T$ time points ($\tilde{x}_{T+1,\ldots,2T}$). By predicting the future steps, model can be trained unsupervised way [23], see Figure (3). We use mean square error ($MSE$) of prediction as the loss, and the adam [24] optimizer for updating the parameter with learning rate 0.0005. In Figure (4.b), we can see an example of the predicted sequences, and it shows that unsupervised learning of the residual D-net learns the dynamic behavior of the human brain.

After unsupervised training, we take all the output of the last layer of RES_cLSTM block ($z_t^{last}$) for classification task. During the classification learning, the parameters in the contracting path($2\times$(RES_D_BLOCK + RES_cLSTM)) can be fine-tuned with concatenated softmax-classifier such as Figure (3.b). And the final decision will be made by averaging the result from classifier as follows:

$$logit = \frac{1}{T} \sum_{t=1}^{N} \text{softmax}(w_{cl} \times z_t^{last} + b_{cl}). \tag{5}$$

Here, $w_{cl}$ and $b_{cl}$ are the softmax-classifier projection weights and bias, respectively. We use the binary cross-entropy as a loss function to fine-tune the architecture with a learning rate 0.00001. We found that involving unsupervised pre-training is crucial, in order to avoid over-fitting during the training of the networks, see Figure (5). After the fine-tuning, the classifier learns the differences between the two dynamic pattern in each class.

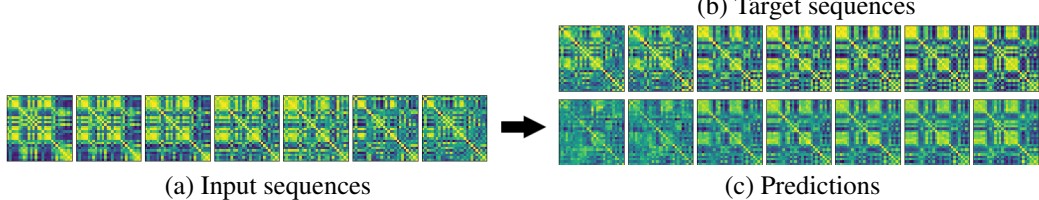

(a) Input sequences          (c) Predictions

Figure 4: (a) shows the sequences of the dynamic functional connectivity that used for input, and (b) shows the target sequences to be predicted and (c) represents the sequences of the predictions from residual D-net.

## 4 Performance Evaluation

We conducted two classification experiments (NC vs. eMCI and NC vs. LMCI) to evaluate the proposed residual D-net and compare it with three baselines techniques. For this, we performed a five-fold subject-wise cross-validation to avoid using the same subject. Each validation set was used for selecting the optimal hyper-parameters for the classification model. The performance was measured by the total accuracy, precision, and recall on the test set.

Table 2: Values of the Accuracy (Acc), Precision (Pre) and Recall (Rec) for each five-fold subject-wise cross-validation for the eMCI dataset.

| CV | SFC+SVM | | | DFC+SVM | | | DAE+HMM | | | Res. D-net | | |
|---|---|---|---|---|---|---|---|---|---|---|---|---|
| | Acc | Pre | Rec | Acc | Pre | Rec | Acc | Pre | Rec | Acc | Pre | Rec |
| 1 | 57.1 | 52.0 | 68.4 | 50.0 | 46.2 | 63.2 | 59.5 | 54.5 | 63.2 | **71.4** | 62.1 | 94.7 |
| 2 | 52.5 | 61.1 | 47.8 | 42.5 | 50.0 | 30.4 | 45.0 | 53.8 | 30.4 | **70.0** | 66.7 | 95.7 |
| 3 | 27.8 | 36.8 | 33.3 | 63.9 | 78.6 | 52.4 | 63.9 | 65.4 | 81.0 | **72.2** | 72.0 | 85.7 |
| 4 | 50.0 | 48.0 | 57.1 | 43.2 | 40.0 | 38.1 | 43.2 | 41.7 | 47.6 | **72.7** | 66.7 | 85.7 |
| 5 | 36.8 | 33.3 | 50.0 | 42.1 | 38.5 | 62.5 | 52.6 | 45.5 | 62.5 | **65.8** | 56.0 | 87.5 |
| Total | 45.5 | 45.9 | 51.0 | 48.0 | 48.0 | 48.0 | 52.5 | 52.3 | 56.0 | **70.5** | 64.7 | 90.0 |

Table 3: Values of the Accuracy (Acc), Precision (Pre) and Recall (Rec) for each five-fold subject-wise cross-validation for the LMCI dataset.

| CV | SFC+SVM | | | DFC+SVM | | | DAE+HMM | | | Res. D-net | | |
|---|---|---|---|---|---|---|---|---|---|---|---|---|
| | Acc | Pre | Rec | Acc | Pre | Rec | Acc | Pre | Rec | Acc | Pre | Rec |
| 1 | 50.0 | 44.4 | 42.1 | 50.0 | 41.7 | 26.3 | 38.1 | 37.9 | 57.9 | **73.8** | 68.2 | 78.9 |
| 2 | 48.5 | 44.4 | 25.0 | 54.5 | 55.6 | 31.3 | 60.6 | 80.0 | 25.0 | **75.8** | 75.0 | 75.0 |
| 3 | **74.1** | 85.7 | 50.0 | 33.3 | 25.0 | 25.0 | 51.9 | 46.7 | 58.3 | 66.7 | 60.0 | 75.0 |
| 4 | 61.1 | 47.1 | 61.5 | 50.0 | 27.3 | 23.1 | 61.1 | 46.7 | 53.8 | **72.2** | 61.5 | 61.5 |
| 5 | 48.7 | 40.0 | 35.3 | 61.5 | 57.1 | 47.1 | 56.4 | 50.0 | 52.9 | **64.1** | 55.6 | 88.2 |
| Total | 55.4 | 48.5 | 41.6 | 50.8 | 41.4 | 31.2 | 53.1 | 46.3 | 49.4 | **70.6** | 63.4 | 76.6 |

## 4.1 Baselines

**Static Functional Connectivity (SFC) + SVM:** Zhang *et al.* [5] suggest five specific pairs of the Pearson's correlation coefficients on each raw dataset ($\mathbf{R} \in \mathbb{R}^{90 \times 130}$), assuming that the FC can be used to distinguish the MCI subjects from the NC. The authors explicitly selected these features after applying a two-sample T-test on 40 subjects.

In this paper, we further investigated twenty alternative coefficients using Fisher feature selection [25], and we fed the selected features to a linear Support Vector Machine (SVM) classifier to perform the classification task.

**Dynamic Functional Connectivity (DFC) + SVM:** In this experiment, in order to consider the brain dynamics, we used a sliding rectangular window (width: 30 TRs) and a 5 TRs stride to estimate the functional connectivity maps $\mathbf{\Sigma}(w) \in \mathbb{R}^{90 \times 90}$ in each window($w = 1, \ldots, 20$). Then, according to [26], we project our data into a $K \times 20$-dimensional feature map and then, we selected the best 100 features using Fisher feature selection, and we measured the performance with a linear SVM classifier.

**Deep Auto-Encoder (DAE) + HMM:** Suk *et al.* [7] propose an unsupervised feature learning using a DAE. First, they trained a four-layer DAE (hidden layers: 200-100-50-2) using as an input all the ROIs directly. Afterward, for each specific time instance, they converted the information of all the ROIs (a 116 real vector) into a 2-dimensional feature map. Then, they fit these 2-dimensional feature maps into two Hidden Markov Models (HMM) to model the NC and the MCI classes. Similarly, we implemented this method but using 6 hidden states with 2-mixtures of Gaussian HMM via the Baum-Welch algorithm.

## 4.2 Discussion and Results

As we discussed during the description of the experiment, we adopted a five-fold subject-wise cross-validation, in order to ensure the reliability of the different methods. Table (2) and Table (3) show the results associated with the accuracy, precision and recall obtained for the different methods, for the eMCI and LMCI dataset respectively.

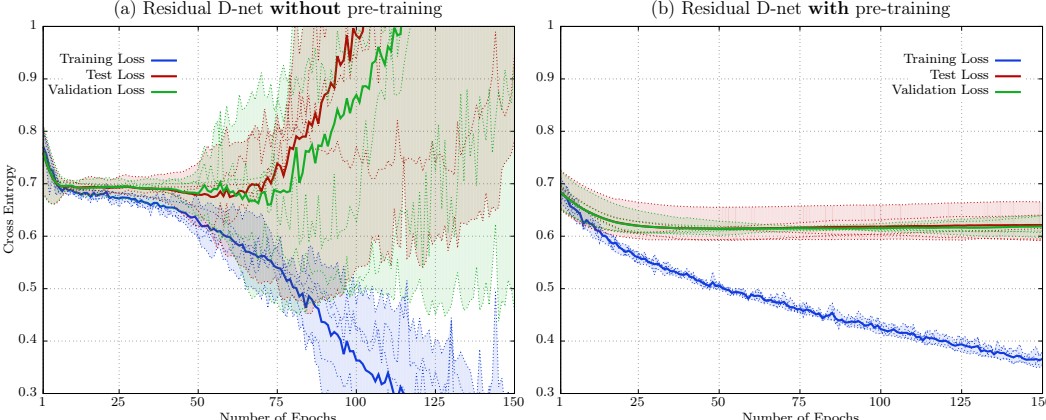

Figure 5: Cross-entropy errors on the LMCI dataset obtained by the proposed method without pre-training (a) and with pre-training (b). The dot lines represent the actual loss errors obtained for each specific cross-validation sets, and the continuous line represents the mean value among all the cross-validation datasets.

The main conclusion is that all the baseline techniques turned out significantly inferior results. First, the inferior performance of SFC+SVM is expected because it does not consider any brain dynamics. Moreover, a further analysis turned out that this method performed well on the training set, in contrast to the test set. This observation evidences that the method fails to generalize among different datasets.

On the other hand, although the DFC+SVM takes into account the time evolution of the FC, the method does not learn the relationships within the brain dynamics and, consequently, fails to perform the classification task.

Regarding to the DAE+HMM, the major limitation of this approach is that is not an end-to-end learning method. That is, although it incorporates an HMM that tries to model the dynamics, the DAE does not capture any information from the brain connectivity dynamics. Leading to a inferior performance.

In contrast, further analysis during the training and the pre-training have shown that our proposed method effectively learns the brain dynamics. Thus, Figure (4) shows the original and the predicted covariance matrices, which assembles the FC brain dynamics. Observe that our proposed approach captures and reproduces the true dynamics of the brain behavior.

This explains why the proposed method exhibits the best performance and it properly generalizes among the different cross-validation sets.

**Pre-training vs. Overfitting**

Considering the limited number of samples of the studied datasets, the primary risk of our proposed method is that of overfitting. However, we faced this challenge by introducing the residual D-net architecture, and also by pre-training the model prior to the classification task.

Although we have already discussed the advantages the residual D-net architecture, we illustrate the benefits of the pre-training in Figure (5), where we plotted the loss errors for the LMCI dataset with and without pre-training.

Observe that the model overfits without pre-training (see Figure (5.a)); that is, we can not guarantee that the method had generalized correctly, making it impossible to establish any proper stopping criterion.

However, the behavior of the loss curves radically changes after pre-training the model (see Figure (5.b)). Now, the method has converged and we can define a proper stopping criterion.

Figure (5) only shows the results for the LMCI dataset, but we have observed the same effects in the eMCI dataset as well.

## 5 Conclusions

In this paper, we presented a new method named residual D-net to identify MCI from NC subjects. In contrast to the previous methods, proposed residual D-net can be efficiently trained with few number of training samples, while unravels the brain connectivity dynamics in unsupervised learning. Furthermore, the proposed pre-training approach robustifies the generalization performance of the proposed method and offers an adequate selection of a stopping criterion in practice.

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
