# OpenReview forum: "Unsupervised learning of the brain connectivity dynamic using residual D-net"
_MIDL.amsterdam/2018/Conference — Submitted to MIDL 2018_

### Review · AnonReviewer1 · 2018-05-04
**Convolutional LSTM to analyze spatio-temporal medical data**

**Rating:** 1
**Confidence:** 2

**Review:**

The paper introduces a D-network designed to learn connectivity pattern in rs-fMRI.

Pros:
+interesting medical application
Cons:
-model design is arbitrary
-week evaluation

I like the idea of using Convolutional LSTM to analyze spatio-temporal medical data. However, the paper's presentation could be improved. It is not clear why the authors call their method unsupervised. From abstract, it is seems that the method is evaluated on a classification task. Moreover, in Figure 1, there are two outputs: 1) prediction of the next time step and 2) classifier. The setup of D-net resembles more a semi supervised learning than unsupervised learning.  Thus, it seems more natural to present D-Net as a semi supervised approach (or maybe even supervised if we have label for each spatiotemporal datapoint).

It looks like D-net is trained in two stages (section 3.3). In the first stage, the model is trained only with prediction loss (pre-training) and, in the second stage, the model is fine-tuned with classification loss. Wouldn't it be possible to train the model in one stage?

The architecture design looks quite arbitrary. It is clear why one would want to combine LSTMs with Convolutional layers; however, the justification on why the D-net would work well is missing. There is a significant body of papers on architectures combining RNNs and convolutional layers. Could the authors comment on why they do not apply previously developed architectures and present a new one? What is the advantage of D-Net when compared to more standard models combining LSTMs and Convolutions?

In Figure 1, what is the meaning of rectangles? Why do we need to copy and add z^{middle}? There is no need to introduce Res_U_Block, same architecture could be explained with a single resblock together with upsampling and downsampling operations.

The text in Figure 3 is too small.

"We found that involving unsupervised pre-training is crucial, in order to avoid over-fitting during the training of the networks ,..." The model pre-training play a role of model regularization. The overfitting depicted in Figure 5, might be due to the lack of proper model regularization.  Did the authors tried to improve regularization of the supervised model by using different regularization techniques: dropout, data augmentation and weight decay? Moreover, from Figure 5(b) we can see that the model with pre-training still gets to the overfitting regime. Further hyper-parameters tuning might be beneficial.

Figure 5 suggests that the authors monitor test set during the experiments. In standard ML approach, one would use the test set only to report the result for the model with the best score on the validation set. If the test set is used to drive design decisions, the reported results might reflect overfitting to test set.

It is not clear where the name D-net is coming from.

There are authors missing in ref 3 and 15.

**Special Issue:**

No

---

> ### Comment · ~Youngjoo_Seo1 · 2018-05-15
> **Response to AnonReviewer1**
>
> Thank you for your feedback.
>
> 1) confusing/misleading titles using "unsupervised learning" and "D-net".
> We really noticed that this is really misleading and confusing word that needs to be revised in a clear way. We want to clarify that the proposed method is trained by 1) unsupervised pre-training and 2) fine-tuning with label information. Therefore, the task itself was solved by supervised manner. We just wanted to emphasize that unsupervised pre-training is really important.
> As all the reviewers pointed out the problem of using the words "unsupervised" and "D-net". Please have a look my answer on "Response to AnonReviewer3 ".
>
> 2) Train the model in one stage
> Yes. It is possible. We tried updating the parameter using two kinds of loss function; predicting sequence loss + classification loss. However, the model affects more on classification loss so that it eventually overfits to the training data. We also trained the model by sparsely updating on classification loss to solve the overfitting problem, but it still has lower performance than training separately.
> It is hard to control the role of each loss, so we decided to train the model separately.
>
> 3) Need to compare with other architecture/ main advantage of D-net
> I really agree with this point. we will put the comparisons on LSTM and convolutional LSTM in the paper.  We had the same question about that so please have a look my answer on "Response to AnonReviewer2" about question 4)
>
> 4) Meaning of rectangles in Fig1.
> It represents the input and output featuremap for each block.
>
> 5) Copy and add z^{middle}
> unlike Res_D_Block, Res_U_Block uses the featuremap from upper-layer(z^{L=1(middle)} and concatenates it with featuremap from low-resolution. This is the skip-connection in U-net architecture that aims to use multi-scale representation. If we don't use the featuremap(z^{L=1(middle)}), the RES_U_BLOCK only considers the low-resolution featuremap. Therefore, we differentiate RES_D/U_BLOCK to emphasize U_block takes z^{L=1(middle).
>
> 6) small text in Fig3.
> We will fix and make them bigger.
>
> 7) Another regularization technique
> Yes! we tried dropout and recurrent batch normalization, but it did not give some significant improvement in the performance. We didn't try data augmentation, but it is really nice idea to try and see that we can have further improvement.
>
> 8) Hyper-parameter tuning
> Yes! it definitely helps. We only search the hyper-parameter of convolution filter size [3 by 3, 4 by 4] and the number of channels starting 16 and 32.
>
> 9) Decision making
> In Fig5, we just plot the loss to show that the model overfits or not. We did not monitor the test loss/acc when we decide to stop training and decision making. We set the parameter that we had the best performance on validation dataset and tested with same parameter.
>
> 10) typos and missing authors in reference.
> Sorry for all the typos and mistakes. We will revise the paper and correct them all.

---

### Review · AnonReviewer2 · 2018-05-09
**Interesting idea to use cLSTM to learn from dynamic FC matrices. Confusing title, overkill naming, insufficient experiments**

**Rating:** 3
**Confidence:** 3

**Review:**

This paper presents a pipeline to diagnose AD/MCI using fMRI images using deep convolutional LSTMs. The network inputs are 28x28 dynamic functional connectivity matrices obtained from 4D fMRI scans. The authors used residual blocks in both the convolutional module and the LSTMs. The proposed methods were evaluated on the fMRI images released by the ADNI cohort. The authors showed the proposed pipeline could outperform several previous solutions that were reproduced by the authors.

pros:
The idea of using convolutional LSTMs to learn from fMRI FC networks is interesting and inspiring. The preprocessing pipeline in this paper makes sense. It is also interesting to see that pre-training by predicting the following series could prevent overfitting after a certain number of iterations. The paper was written in a clear way.

cons:
The title is inaccurate and confusing. The main motivation of this paper is to diagnose AD/MCI as a supervised task. The unsupervised learning was only used for pre-training the network. Besides Fig.5, it is not clear if the unsupervised learning is really necessary here since the best test performance without the pre-training stage is not shown. Is it replaceable by the other regularization methods? eg. early stopping, dropout, batch-normalization, etc. Experimental evidence is needed to make such claim. It also remains unclear which choice of the architecture design is critical without ablative analysis made available.

The naming of D-Net is indeed an overkill and non-necessary. It seems it is called D-Net only because of the Residual blocks. Then why bother naming it?

The evaluation falls short of a comprehensive analysis of the claims made by this paper. The paper claims the advantages of using residual blocks in cLSTMs which also resulted in the network naming. However, the numbers from a cLSTM without the residual blocks were not provided. It is intuitive nowadays that deep learning methods could outperform the hand-crafted features. It would be more interesting to see the results from other simpler deep learning approaches, for example, the performance of a 3D CNN or a CNN with anisotropic kernels.

The dataset used in this study is way too small (36+31+26 patients), to draw any solid conclusions in a machine learning study regarding the classification accuracy.

**Special Issue:**

Yes

---

> ### Comment · ~Youngjoo_Seo1 · 2018-05-15
> **Response to AnonReviewer2**
>
> Thank you for your feedback.
>
> 1) confusing/misleading titles using "unsupervised learning" and "D-net".
> Yes. this is definitely revised not to make readers confusing. This is common comments that I've got. So please have a look my answer on "Response to AnonReviewer3 ".
>
> 2) Missing best test performance on without pre-training.
> In Fig5, we show that the cross-entropy loss along with training iteration. Also, we have the similar graph on training/test/validation accuracy both with/without pre-training(which we didn't put in the paper). We can also put the test performance on without pretraining on table 2 and 3.
> The main reason why we don't put the best performance of "without pre-training" is that the performance really overfits to training, and we can not trust the results. The performance is changing by random initialization of the parameter(random seed).
>
> 3) Another regularization methods
> We tried the dropout when we do the pre-training, but it does not change the performance so much. Also, we tried recurrent batch-normalization[Tim Cooijmans 2016], but didn't get significant improvement. So we decided to focus on unsupervised pre-training that we want to emphasize. However, we will further study on other regularization technique for better performance.
>
> 4) Which choice of the architecture design is critical
> This is an important point to represents why we design the residual D-net. Actually, we tried simple LSTM on raw data(without using correlation map). But the method is overfitting though we tried unsupervised pre-training. That's the reason why we think further to use correlation map as an input.
> Then, we also tried convolutional LSTM using correlation map. But it does not predict the future sequences well. That's the reason why we move to think multi-scale expression by combining U-net architecture.
> We will put the result of other architecture for clarification.
>
> 5) Small dataset.
> That is the limitation of the study on fMRI dataset. We are thinking to increase the Healthy(NC) dataset that we can have from other data pool for future study.

---

### Review · AnonReviewer3 · 2018-05-10
**Unclear model description and small evaluation data**

**Rating:** 2
**Confidence:** 2

**Review:**

The authors adopt a multiscale convolutional LSTM to classify mild cognitive impaired (MCI) from normal healthy control (NC) subjects. The model takes a sequence of correlation matrices as inputs that are intended to approximate brain dynamics. Before the classification stage, the model is pretrained using an LSTM unit. At each time step, the model predicts the correlation matrix of the next time step. The authors also incorporate multiscale information in their model by using tailored up- and downscaling layers. The proposed model is evaluated against three approaches from the literature.

The achieved results of the model look encouraging but I have some concerns with the clarity, evaluation, and maturity of the manuscript.

* The title of the paper should reflect that the proposed model is a convolutional LSTM with pretraining, not unsupervised learning. Why do the authors call this model a D-net? Skip-connections do not justify a new model name in my point of view.

* The description of the architecture is interspersed with parameter settings, e.g., 3x3 kernel while explaining the layer operations. I would recommend to separate the architectural description from the parameterization of the model.

* Figure 1 seems overloaded and so I am wondering if it can be broken down in more parts. What are the dashed green, blue, black, and red arrows? What does the copy&add transformation mean and why is it important?

* The evaluation dataset is quite small (277 scans from 93 patients) which makes it hard to assess the performance of the model. What are the confidence intervals for the reported results? How many hyperparameters have been tried out?

* The manuscript contains many grammar mistakes and typos. I recommend to carefully proofread the manuscript.

* The notation is in some parts confusing or incomplete. For instance, a mix of superscripts (l=0, middle, last) is used.

**Special Issue:**

No

---

> ### Comment · ~Youngjoo_Seo1 · 2018-05-15
> **Response to AnonReviewer3**
>
> Thanks for your feedback.
>
> 1) confusing/misleading titles using "unsupervised learning" and "D-net".
> We agree that the words on titles are misleading the paper. The model was trained by unsupervised pretraining and fine-tuning with label information. We just want to emphasize the importance of the unsupervised pretraining to overcomes the overfitting problem on classification task. Also, the expression of "D-net" can be replaced with "recurrent U-net" for easy understanding. Actually, D-shapes comes from figure1. that we use U-net structure to be rotated 90' to express recurrent flow together.
>
> 2) Description of the architecture on Figure1.
> The rectangles are input and output featuremap for each block. The numbers inside the rectangle box is [width, height, channel] of featuremap. And we want to clarify that we use "three depth layer(L=0, 1(middle), 2(last)) ". We will revise and change the notation "middle" and "last" into L=1, L=2. (Here, I use upper-case L instead of lower-case l for easy reading)
>
> The dashed green line represents the recurrent flow, that residual convolution lstm block takes the previous hidden state(h_{t-1}^L_1, h_{t-1}^L_2) to generate current time hidden state(h_{t}^L_1, h_{t}^L_2). And the red line with "Copy & Add" is the skip-connection to feed the output of middle layer(L=1) to RES_U_BLOCK. In the RES_U_BLOCK, this featuremap are concatenated with up-conv featuremap(hat_y_t^L) that represented in equation 2) as [hat_y_t^L, z_t^L]. So, the z_t^L=1(middle) is [14, 14, 16] featuremap, and hat_y_t^L=1 is [14, 14, 16]. so the concatenation is [14, 14, 32] featuremap.
> The dashed black line is the expression that the featuremap of the last layer is used for input of classifier.
>
> 3) Small dataset
> Yes. We agree that the dataset is quite small to train the model by just supervised learning. Therefore, we first pre-trained the model by predicting the future steps. In the pre-training phase, the model uses 277 samples and each sample has 56 sequences of correlation map. By predicting the half(23) sequences of future step, it is possible to train the model parameter. You can see how the unsupervised pre-training works well in Figure4.
> After the pre-training is done, we can achieve stable performance on test dataset. We will put the figure of accuracy per each iteration to show that the performance is confidence and stable.
>
> 4) grammar mistakes and confusing notation.
> Sorry for all the typos and mistakes. We will revise the paper and correct them.

---

### Decision · Program_Chairs · 2018-05-15
**Paper80 Acceptance Decision**

Reject